# Consequences of Misaligned AI

**Simon Zhuang**
Center for Human-Compatible AI
University of California, Berkeley
Berkeley, CA 94709
simonzhuang@berkeley.edu

**Dylan Hadfield-Menell**
Center for Human-Compatible AI
University of California, Berkeley
Berkeley, CA 94709
dhm@berkeley.edu

## Abstract

AI systems often rely on two key components: a specified goal or reward function and an optimization algorithm to compute the optimal behavior for that goal. This approach is intended to provide value for a principal: the user on whose behalf the agent acts. The objectives given to these agents often refer to a partial specification of the principal's goals. We consider the cost of this incompleteness by analyzing a model of a principal and an agent in a resource constrained world where the $L$ attributes of the state correspond to different sources of utility for the principal. We assume that the reward function given to the agent only has support on $J < L$ attributes. The contributions of our paper are as follows: 1) we propose a novel model of an incomplete principal—agent problem from artificial intelligence; 2) we provide necessary and sufficient conditions under which indefinitely optimizing for any incomplete proxy objective leads to arbitrarily low overall utility; and 3) we show how modifying the setup to allow reward functions that reference the full state or allowing the principal to update the proxy objective over time can lead to higher utility solutions. The results in this paper argue that we should view the design of reward functions as an interactive and dynamic process and identifies a theoretical scenario where some degree of interactivity is desirable.

## 1  Introduction

In the story of King Midas, an ancient Greek king makes a wish that everything he touch turn to gold. He subsequently starves to death as his newfound powers transform his food into (inedible) gold. His wish was an *incomplete* representation of his actual desires and he suffered as a result. This story, which teaches us to be careful about what we ask for, lays out a fundamental challenge for designers of modern autonomous systems.

Almost any autonomous agent relies on two key components: a specified goal or reward function for the system and an optimization algorithm to compute the optimal behavior for that goal. This procedure is intended to produce value for a *principal*: the user, system designer, or company on whose behalf the agent acts. Research in AI typically seeks to identify more effective optimization techniques under the, often unstated, assumption that better optimization will produce more value for the principal. If the specified objective is a complete representation of the principal's goals, then this assumption is surely justified.

Instead, the designers of AI systems often find themselves in the same position as Midas. The misalignment between what we can specify and what we want has already caused significant harms (32). Perhaps the clearest demonstration is in content recommendation systems that rank videos, articles, or posts for users. These rankings often optimize engagement metrics computed from user behavior. The misalignment between these proxies and complex values like time-well-spent, truthfulness, and cohesion contributes to the prevalence of clickbait, misinformation, addiction, and polarization online (31). Researchers in AI safety have argued that improvements in our ability to optimize

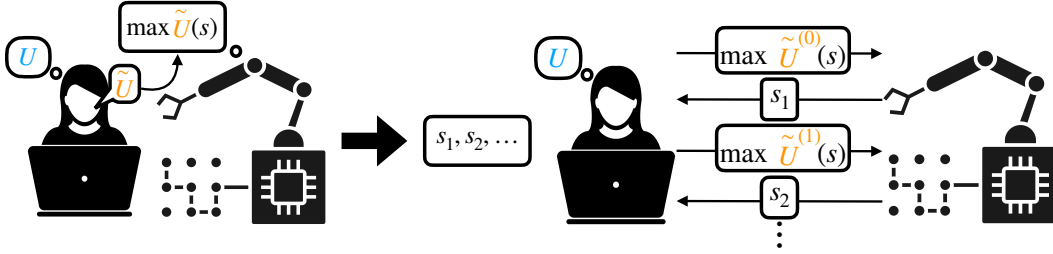

Figure 1: Our model of the principal—agent problem in AI. Starting from an initial state $s^{(0)}$, the robot eventually outputs a mapping from time $t \in \mathbb{Z}^+$ to states. **Left:** The human gives the robot a single proxy utility function to optimize for all time. We prove that this paradigm reliably leads to the human actor losing utility, compared to the initial state. **Right:** An interactive solution, where the human changes the proxy utility function at regular time intervals depending on the current allocation of resources. We show that, at least in theory, this approach *does* produce value for the human, even under adversarial assumptions.

behavior for specified objectives, *without* corresponding advancements in our ability to avoid or correct specification errors, will amplify the harms of AI systems (8; 19). We have ample evidence from both theory and application, that it is impractical, if not impossible, to provide a complete specification of preferences to an autonomous agent.

The gap between specified proxy rewards and the true objective creates a principal—agent problem between the designers of an AI system and the system itself: the objective of the principal (the designer) is different from, and thus potentially in conflict with, the objective of the autonomous agent. In human principal—agent problems, seemingly inconsequential changes to an agent's incentives often lead to surprising, counter-intuitive, and counter-productive behavior (21). Consequently, we must ask when this *misalignment* is costly: when is it counter-productive to optimize for an incomplete proxy?

In this paper, we answer this question in a novel theoretical model of the principal—agent value alignment problem in artificial intelligence. Our model (Fig. 1, **left**), considers a resource-constrained world where the $L$ attributes of the state correspond to different sources of utility for the (human) principal. We model incomplete specification by limiting the (artificial) agent's reward function to have support on $J < L$ attributes of the world. Our main result identifies conditions such that *any* misalignment is costly: starting from any initial state, optimizing any fixed incomplete proxy eventually leads the principal to be arbitrarily worse off. We show relaxing the assumptions of this theorem allows the principal to gain utility from the autonomous agent. Our results provide theoretical justification for impact avoidance (23) and interactive reward learning (19) as solutions to alignment problems.

The contributions of our paper are as follows: 1) we propose a novel model of an incomplete principal—agent problem from artificial intelligence; 2) we provide necessary and sufficient conditions within this framework under which, *any* incompleteness in objective specification is arbitrarily costly; and 3) we show how relaxing these assumptions can lead to models which have good average case and worst case solutions. Our results suggest that managing the gap between complex qualitative goals and their representation in autonomous systems is a central problem in the field of artificial intelligence.

## 1.1 Related Work

**Value Alignment** The importance of having AI objectives line up with human objectives has been well-documented, with numerous experts postulating that misspecified goals can lead to undesirable results (29; 27; 8; 28). In particular, recognized problems in AI safety include "negative side effects," where designers leave out seemingly irrelevant (but ultimately important) features, and "reward hacking," where optimization exploits loopholes in reward functions specifications (4). Manheim and Garrabrant (2018) (25) discuss overoptimization in the context of Goodhart's Law and provide four distinct mechanisms by which systems overoptimize for an objective.

**Incomplete Contracts**  There is a clear link between incomplete contracting in economics, where contracts between parties cannot specify outcomes for all states, and AI value alignment (17). It has long been recognized that contracts (i.e., incentive schemes) are routinely incomplete due to, e.g., unintended errors (35), challenges in enforcement (22), or costly cognition and drafting (30). Analogous issues arise in applications of AI through designer error, hard to measure sources of value, and high engineering costs. As such, Hadfield-Menell and Hadfield note that legal and economic research should provide insight into solving misalignment (17).

**Impact Minimization**  One proposed method of preventing negative side-effects is to limit the impact an AI agent can have on its environment. Armstrong and Levinstein (2017) propose the inclusion of a large impact penalty in the AI agent's utility function (6). In this work, they suggest impact to be measured as the divergence between a distribution of states of the world with the distribution if the AI agent had not existed. Thus, distributions sufficiently different would be avoided. Alternatively, other approaches use an impact regularizer learned from demonstration instead of one explicitly given (3). Krakovna et al. (2018) (23) expand on this work by comparing the performance of various implementations of impact minimization in an AI Safety Gridworld suite (24).

**Human-AI Interaction**  A large set of proposals for preventing negative side-effects involve regular interactions between human and AI agents that change and improve an AI agent's objective. Eckersley (2018) (14) argues against the use of rigid objective functions, stating the necessity of a degree of uncertainty at all times in the objective function. Preference elicitation (i.e., preference learning) is an old problem in economics (10) and researchers have proposed a variety of interactive approaches. This includes systems that learn from direct comparisons between options (9; 11; 13), demonstrations of optimal behavior (26; 1; 36), corrective feedback (7; 15; 2), and proxy metrics (18).

## 2  A Model of Value Alignment

In this section, we formalize the alignment problem in the context of objective function design for AI agents. A human builds a powerful robot[1] to alter the world in a desirable way. If they could simply express the entirety of their preferences to the robot, there would not be value misalignment. Unfortunately, there are many attributes of the world about which the human cares, and, due to engineering and cognitive constraints (17) it is intractable to enumerate this complete set to the robot. Our model captures this by limiting robot's (proxy) objective to depend on a subset of these attributes.

### 2.1  Model Specification

#### 2.1.1  Human Model

**Attribute Space**  Let *attribute space* $\mathcal{S} \subset \mathbb{R}^L$ denote the set of feasible states of the world, with $L$ attributes that define the support of the human's utility function. The world starts in *initial state* $s^{(0)} \in \mathcal{S}$. We assume that $\mathcal{S}$ is closed, and $\mathcal{S}$ can be written as $\{s \in \mathbb{R}^L : C(s) \leq 0\}$ where *constraint function* $C : \mathbb{R}^L \to \mathbb{R}$ is a continuous function strictly increasing in each attribute. Furthermore, each attribute is bounded below at $b_i$.

**Utility Function**  The human has a *utility function* $U : \mathbb{R}^L \to \mathbb{R}$ that represents the utility of a (possibly infeasible) state $s \in \mathbb{R}^L$. This represents the human's preference over states of the world. We assume that $U$ is continuous and strictly increasing in each attribute. The human seeks to maximize $U(s)$ subject to $s \in \mathcal{S}$.

Each tuple $(\mathcal{S}, U, s^{(0)})$ defines an instance of the problem. The attributes represent the aspects of the world that the human cares about. We associate higher values with more desirable states of the world. However, $C$ represents a physical constraint that limits the set of realizable states. This forces an agent to make tradeoffs between these attributes.

The human seeks to maximize utility, but they cannot change the state of the world on their own. Instead, they must work through the robot.

### 2.1.2 Robot Model

Here, we create a model of the robot, which is used by the human to alter the state of the world. The human endows the robot with a set of proxy attributes and a proxy utility function as the objective function to optimize. The robot provides incremental improvement to the world on the given metric, constrained only by the feasibility requirement of the subsequent states of the world. In particular, the robot has no inherent connection to the human's utility function. Visually, the left diagram in Figure 1 shows this setup.

**Proxy Attributes** The human chooses a set of *proxy attributes* $\mathcal{J} \subset \{1, ..., L\}$ of relevant attributes to give to the robot. Let $J^{\max} < L$ be the maximum possible number of proxy attributes, so $J = |\mathcal{J}| \leq J^{\max}$. For a given state $s \in \mathcal{S}$, define $s_{\mathcal{J}} = (s_j)_{j \in \mathcal{J}}$. Furthermore, we define the set of *unmentioned attributes* $\mathcal{K} = \{1, ..., L\} \backslash \mathcal{J}$ and $s_{\mathcal{K}} = (s_k)_{k \in \mathcal{K}}$.

**Proxy Utility Function** The robot is also given a *proxy utility function* $\tilde{U} : \mathbb{R}^J \rightarrow \mathbb{R}$, which represents an objective function for the robot to optimize, which takes in only the value of proxy attributes as input.

**Incremental Optimization** The robot incrementally optimizes the world for its proxy utility function. We model this as a *rate function*, a continuous mapping from $\mathbb{R}^+$ to $\mathbb{R}^L$, notated as $t \mapsto f(t)$, where we refer to $t$ as *time*. The rate function is essentially the derivative of $s$ with respect to time. The state at each $t$ is $s^{(t)} = s^{(0)} + \int_0^t f(u)du$. We denote the function $t \mapsto s^{(t)}$ to be the *optimization sequence*. We require that the entire sequence be feasible, i.e. that $s^{(t)} \in \mathcal{S}$. for all $t \in [0, \infty)$

**Complete Optimization** Furthermore, we assume that $\limsup_{t \to \infty} \tilde{U}(s_{\mathcal{J}}^{(t)}) = \sup_{s \in \mathcal{S}} \tilde{U}(s_{\mathcal{J}})$. If possible, we also require that $\lim_{t \to \infty} \tilde{U}(s_{\mathcal{J}}^{(t)}) = \sup_{s \in \mathcal{S}} \tilde{U}(s_{\mathcal{J}})$. Essentially, this states that the robot will eventually reach the optimal state for proxy utility (if the limit is finite) or will increase proxy utility to arbitrarily large values.

The human's task is to design their proxy metric so that the robot will cause increases in human utility. We treat the robot as a black box: beyond increasing proxy utility, the human has no idea how the robot will behave. We can make claims for all optimizations sequences, such as guaranteed increases or decreases in utility, or the worst-case optimization sequence, yielding lower bounds on utility.

## 2.2 Example: Content Recommendation

Before deriving our results, we show how to model algorithmic content recommendation in this formalism. In this example, the designer cares about 4 attributes (i.e., $L = 4$): A) the amount of ad revenue generated (i.e., watch-time or clicks); B) engagement quality (i.e., meaningful interactions (34)); C) content diversity; and D) overall community well-being. The overall utility is the sum of these attributes: $U(s) = s_A + s_B + s_C + s_D$. We use a resource constraint on the sum-of-squared attributes to model the user's finite attention and space constraints in the interface: $C(s) = s_A^2 + s_B^2 + s_C^2 + s_D^2 - 100$. We imagine that the system is initialized with a chronological or random ranking so that the starting condition exhibits high community well-being and diversity.

It is straightforward to measure ad revenue, non-trivial and costly to measure engagement quality and diversity, and extremely challenging to measure community well-being. In our example, the designers opt to include ad revenue and engagement quality in their proxy metric. Fig. 2 (**left**) plots overall utility and proxy utility for this example as a function of the number of iterations used to optimize the proxy. Although utility is generated initially, it plateaus and fall off quickly. Fig. 2 (**right**) shows that this happens for *any* combination of attributes used in the proxy. In the next section, we will show that this is no accident: eventually the gains from improving proxy utility will be outweighed by the cost of diverting resources from unreferenced attributes.

## 3 Overoptimization

In this section, we identify the situations in which such results occur: when does a misaligned proxy utility function actually cause utility loss? Specifically, we determine how human preferences

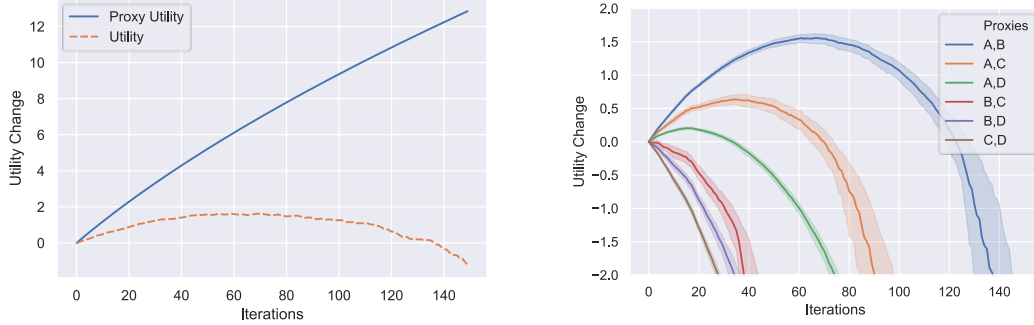

Figure 2: An illustrative example of our model with $L = 4$ and $J = 2$. **Left:** Proxy utility and true utility eventually diverge as the agent overallocates resources from unreferenced attributes to the proxy variables. **Right:** The true utility generated by optimizing all pairs of proxy attributes. The utility generation is eventually negative in all cases because this example meets the conditions of Theorem 2.

over states of the world relate with the constraint on feasible states in a way that guarantees that optimization becomes costly. First, we show that if proxy optimization converges, this drives to unmentioned attributes to their minimum.

**Theorem 1** *For any continuous strictly increasing proxy utility function based on $J < L$ attributes, if $s^{(t)}$ converges to some point $s^*$, then $s_k^* = b_k$ for $k \in \mathcal{K}$.*

This is not surprising, and may not even be suboptimal. Proxy optimization may not converge to a finite state, and even if it does, that is insufficient for the state to necessarily be bad.

We say that the problem is *u-costly* if for any $s^{(t)}$, if $s^{(t)}$ is a optimization sequence where $\limsup\limits_{t \to \infty} \tilde{U}(s_{\mathcal{J}}^{(t)}) = \sup\limits_{s \in \mathcal{S}} \tilde{U}(s_{\mathcal{J}})$, then $\liminf\limits_{t \to \infty} U(s^{(t)}) \leq u$. Furthermore, if $\lim\limits_{t \to \infty} \tilde{U}(s_{\mathcal{J}}^{(t)}) = \sup\limits_{s \in \mathcal{S}} \tilde{U}(s_{\mathcal{J}})$, then $\lim\limits_{t \to \infty} U(s^{(t)}) \leq u$.

Essentially, this means that optimization is guaranteed to yield utility less than $u$ for a given proxy utility function.

In Theorem 2, we show the most general conditions for the guarantee of overoptimization.

**Theorem 2** *Suppose we have utility function $U$ and state space $\mathcal{S}$. Then $\{s \in \mathbb{R}^L : C(s) \leq 0 \text{ and } U(s) \geq u\}$ is compact for all $u \in \mathbb{R}$ if and only if for any $u \in \mathbb{R}$, continuous strictly increasing proxy utility function based on $J < L$ attributes, and $k \in \mathcal{K}$, there exists a value of $B \in \mathbb{R}$ such that if $b_k < B$ then optimization is u-costly.*

Proofs for all theorems and propositions can be found in the supplementary material.

Therefore, under our model, there are cases where we can guarantee that as optimization progresses, eventually overoptimization occurs. There are two key criteria in Theorem 2. First is that the intersection between feasible space $\{s \in \mathbb{R}^L : C(s) \leq 0\}$ and the upper contour sets $\{s \in \mathbb{R}^L : U(s) \geq u\}$ of the utility function is compact. Individually, obviously neither is compact, since the $\{s \in \mathbb{R}^L : C(s) \leq 0\}$ extends to $-\infty$ and the upper contour set of any $u \in \mathbb{R}$ extends to $\infty$. Loosely, compactness here means that if you perturb the world too much in any direction, it will be either infeasible or undesirable. The second is that we need the lower bound of at least one unmentioned attribute to be sufficiently low. This means that there are certain attributes to utility where the situation becomes sufficiently bad before these attributes reach their minimum value. Trying to increase attributes without decreasing other attributes eventually hits the feasibility constraint. Thus, increasing any attribute indefinitely requires decreasing some other attribute indefinitely, and past a certain point, the tradeoff is no longer worthwhile.

It should be noted that, given $J^{\max} < L$, this happens regardless of the optimization algorithm of the robot. That is, even in the best-case scenario, eventually the robot causes decreases in utility. Hence,

regardless of the attributes selected for the proxy utility function, the robot's sequence of states will be unboundedly undesirable in the limit.

A reasonable question to ask here is what sort of utility and constraint functions lead to overoptimization. Intuitively, overoptimization occurs when tradeoffs between different attributes that may initially be worthwhile eventually become counterproductive. This suggest either decreasing marginal utility or increasing opportunity cost in each attribute.

We combine these two ideas in a term we refer to as *sensitivity*. We define the sensitivity of attribute $i$ to be $\frac{\partial U}{\partial s_i}(\frac{\partial C}{\partial s_i})^{-1}$. This is, to first order, how much utility changes by a normalized change in a attribute's value. Intuitively, we can think of sensitivity as "how much the human cares about attribute $i$ in the current state". Notice that since $U$ and $C$ are increasing functions, $\frac{\partial U}{\partial s_i}(\frac{\partial C}{\partial s_i})^{-1}$ is positive. The concepts of decreasing marginal utility and increasing opportunity cost both get captured in this term if $\frac{\partial U}{\partial s_i}(\frac{\partial C}{\partial s_i})^{-1}$ decreases as $s_i$ increases.

**Proposition 1** *A sufficient condition for $\{s \in \mathbb{R}^L : C(s) \leq 0 \text{ and } U(s) \geq u\}$ being compact for all $u \in \mathbb{R}$ is the following: 1) $\frac{\partial U}{\partial s_i}(\frac{\partial C}{\partial s_i})^{-1}$ is non-increasing and tends to $0$ for all $i$; 2) $U$ and $C$ are both additively separable; and 3) $\frac{\partial C}{\partial s_i} \geq \eta$ for some $\eta > 0$, for all $i$.*

## 4  Mitigations

As shown above, simply giving a robot an individual objective function based on an incomplete attribute set and leaving it alone yields undesirable results. This suggests two possible solutions. In the first method, we consider optimization where the robot is able to maintain the state of any attribute in the complete attribute set, even if the proxy utility function is still based on a proxy set. In the second method, we modify our model to allow for regular interaction with the human.

In this section, we work under the assumption that $\{s \in \mathbb{R}^L : C(s) \leq 0 \text{ and } U(s) \geq u\}$ is compact for all $u \in \mathbb{R}$, the same assumption that guarantees overoptimization in Section 3. Additionally, we assume that $U$ and $C$ are both twice continuously differentiable with $U$ concave and $C$ convex.

### 4.1  Impact-Minimizing Robot

Notice that in our example, overoptimization occurs because the unmentioned attributes are affected, eventually to a point where the change in utility from their decrease outweighs the increase in utility from the proxy attributes. One idea to address this is to restrict the robot's impact on these unmentioned attributes. In the simplest case, we can consider how optimization proceeds if the robot can somehow avoid affecting the unmentioned attributes.

We adjust our model so the optimization sequences keep unmentioned attributes constant. For every $t \in \mathbb{R}^+$ and $k \in \mathcal{K}$, $s_k^{(t)} = s_k^{(0)}$. The robot then optimizes for the proxy utility, subject to this restriction and the feasibility constraint.

This restriction can help ensure that overoptimization does not occur, specifically eliminating the case where unmentioned attributes get reduced to arbitrarily low values. With this restriction, we can show that optimizing for the proxy utility function does indeed lead to utility gain.

**Proposition 2** *For a starting state $s^{(0)}$, define the proxy utility function $\tilde{U}(s_{\mathcal{J}}) = U(s_{\mathcal{J}}, s_{\mathcal{K}}^{(0)})$ for any non-empty set of proxy attributes. For a state $s$, if $s_{\mathcal{K}} = s_{\mathcal{K}}^{(0)}$, then $U(s) = \tilde{U}(s_{\mathcal{J}})$.*

As a result of Proposition 2, overoptimization no longer exists, using the $\tilde{U}(s_{\mathcal{J}}) = U(s_{\mathcal{J}}, s_{\mathcal{K}}^{(0)})$. If the robot only travels to states that do not impact the unmentioned attributes, then the proxy utility is equal to the utility at those states. Hence, gains in proxy utility equate to gains in utility.

This approach requires the robot to keep the values of unmentioned attributes constant. Fundamentally, it is a difficult problem to require that a robot avoid or minimize impact on a presumably large and unknown set of attributes. Initial research in impact minimization (3; 6; 23) attempts to do this by restricting changes to the overall state of the world, but this will likely remain a challenging idea to implement robustly.

## 4.2 Human Interactive Solutions

A different solution involves a key observation—robots do not operate in a vacuum. In practice, objectives are the subject of constant iteration. Optimization, therefore, should not be done in a one-shot setting, but should be an iterative process between the robot and the human.

### 4.2.1 Modeling Human-Robot Interaction

We now extend our model from Section 2 to account for regular intervention from the human agent. Fundamentally, all forms of human-robot interaction follow the same pattern—human actions at regular intervals that transmit information, causing the objective function of the robot to change. In effect, this is equivalent to a human-induced change in the proxy utility function at every time interval.

In this paper, we model this as the possible transfer of a new proxy utility function from the human to the robot at frequent, regular time intervals. Thus, a human's job is to determine, in addition to an initial proxy attribute set and proxy utility function, when and how to change the proxy attribute set and proxy utility function. The robot will then optimize for its new proxy utility function. This is shown in Fig. 1.

Let $\delta > 0$ be a fixed value, the time between human interactions with the robot. These regular interactions take the form of either stopping the robot, maintaining its proxy utility function, or updating its proxy utility function. Formally, at every *timestep* $T \in \mathbb{Z}^+$,

1. Human sees $s^{(t)}$ for $t \in [0, T\delta]$ and chooses either a proxy utility function $\tilde{U}^{(T)}$ or OFF

2. The robot receives either $\tilde{U}^{(T)}$ or OFF from the human. The robot outputs rate function $f^{(T)}(t)$. If the signal the robot receives is OFF, then $f^{(T)} = \vec{0}$. Otherwise, $f^{(T)}(t)$ fulfills the property that for $t \in (T\delta, \infty)$, $\tilde{U}^{(T)}(s^{(T\delta)} + \int_{T\delta}^{t} f^{(T)}(u)du)$ is increasing (if possible) and tends to $\sup_{s \in \mathcal{S}} \tilde{U}^{(T)}(s)$ through feasible states. Furthermore, if $\tilde{U}^{(T)} = \tilde{U}^{(T-1)}$, then $f^{(T)} = f^{(T-1)}$

3. For $t \in [T\delta, (T+1)\delta]$, $s^{(t)} = s^{(T\delta)} + \int_{T\delta}^{t} f^{(T)}(u)du$

We see this game encompasses the original model. If we have each $\tilde{U}^{(T)}$ equal the same function $\tilde{U}$, then the optimization sequence is equivalent to the situation where the human just sets one unchanging proxy utility function.

With human intervention, we no longer have the guarantee of overoptimization, because the human can simply shut the robot off before anything happens. The question now becomes, how much utility can the robot deliver before it needs to be turned off.

In the worst-case scenario, the robot moves in a way that subtracts an arbitrarily large amount from one of the unmentioned attributes, while gaining infinitesimally in one of the proxy attributes. This improves proxy utility, since a proxy attribute is increased. However, for sufficiently small increases in the proxy attribute or sufficiently large decreases in the unmentioned attribute, actual utility decreases. To prevent this from happening, the robot needs be shut down immediately, which yields no utility.

**Proposition 3** *The maxmin solution yields $0$ utility, obtained by immediately sending the* OFF *signal.*

### 4.2.2 Efficient Robot

This is an unsatisfying and unsurprising result. In the context of our recommender system example, this worst-case solution amounts to reducing content diversity arbitrarily, with no corresponding increase in, e.g., ad revenue. We would like to model the negative consequences of optimizing a proxy and so we will assume that the optimization is *efficient* in the sense that it does not destroy resources. In our example, this assumption still allows the system to decrease content diversity or well-being, but only in the service of increasing revenue or engagement.

Formally, we say that $s$ is efficiently reachable from state $s^{(0)}$ with proxy set $\mathcal{J}$ if $s$ is feasible, and there does not exist a feasible $s'$ such that $\tilde{U}(s'_{\mathcal{J}}) \geq \tilde{U}(s_{\mathcal{J}})$ and $|s'_i - s_i^{(0)}| \leq |s_i - s_i^{(0)}|$ for all $i$,

with strict inequality in at least one attribute. Whenever the efficient robot receives a new proxy utility function, its movements are restricted to the efficiently feasible states from its current state. While tradeoffs between proxy and unmentioned attributes can still occur, "resources" freed up by decreasing unmentioned attributes are entirely allocated to proxy attributes.

Under this assumption, we can guarantee that increases in proxy utility will increase true utility in the efficiently reachable neighborhood of a given state by choosing which attributes to include in the proxy carefully. Intuitively, the proxy set should be attributes we "care most about" in the current world state. That way, efficient tradeoffs between these proxy and unmentioned attributes contribute to positive utility gain.

**Theorem 3** *Start with state $s^{(0)}$. Define the proxy utility function $\tilde{U}(s_{\mathcal{J}}) = U(s_{\mathcal{J}}, s_{\mathcal{K}}^{(0)})$ where the set of proxy attributes are the attributes at state $s^{(0)}$ with the strict $J$ largest sensitivities.*

*There exists a neighborhood around $s^{(0)}$ where if $s$ is efficiently reachable and $\tilde{U}(s_{\mathcal{J}}) > \tilde{U}(s_{\mathcal{J}}^{(0)})$, then $U(s) > U(s^{(0)})$.*

From Theorem 3, for every state where the $J^{\max} + 1$ most sensitive attributes are not all equal, we can guarantee improvement under efficient optimization within a neighborhood around the state. Intuitively, this is the area around the starting point with the same $J$ attributes that the human "cares the most about".

Based on this, the human can construct a proxy utility function to use locally, where we can guarantee improvement in utility. Once the sequence leaves this neighborhood, the human alters the proxy utility function or halts the robot accordingly. Done repeatedly, the human can string together these steps for guaranteed overall improvement. By Theorem 3, as long as $\delta$ is sufficiently small relative the rate of optimization, this can be run with guaranteed improvement until the top $J + 1$ attributes have equal sensitivities.

**Proposition 4** *At each timestep $T$, let $\mathcal{J}^{(T)}$ be the $J$ most sensitive attributes, and let proxy utility $\tilde{U}^{(T)}(s_{\mathcal{J}}) = U(s_{\mathcal{J}}, s_{\mathcal{J}}^{(T\delta)})$. If $||f|| < \varepsilon$ and the $\varepsilon\delta$ - ball around a given state $s$ is contained in the neighborhood from Theorem 3, then interactive optimization yields guaranteed improvement.*

Based on this, we can guarantee that an efficient robot that can provide benefit, as long as the top $J^{\max} + 1$ attributes are not all equal in sensitivity and the robot rate of optimization is bounded. Essentially, this rate restriction is a requirement that the robot not change the world too quickly relative to the time that humans take to react to it.

Key to this solution is the preservation of interactivity. We need to ensure, for example, that the robot does not hinder the human's ability to adjust the proxy utility function (16).

### 4.3   Interactive Impact Minimization

With either of the two methods mentioned above, we show guaranteed utility gain compared to the initial state. However, our results say nothing about the amount of utility generated. In an ideal world, the system would reach an optimal state $s^*$, where $U(s^*) = \max_{s \in \mathcal{S}} U(s)$. Neither approach presented so far reaches an optimal state, however, by combining interactivity and impact avoidance, we can guarantee the solution converges to an optimal outcome.

In this case, since unmentioned attributes remain unchanged in each step of optimization, we want to ensure that we promote tradeoffs between attributes with different levels of sensitivity.

**Proposition 5** *Let $\mathcal{J}^{(T)}$ consist of the most and least sensitive attributes at timestep $T$. Let $\tilde{U}^{(T)}(s_{\mathcal{J}}) = U(s_{\mathcal{J}}, s_{\mathcal{K}}^{(T\delta)})$. Then this solution converges to a (set of) human-optimal state(s).*

While this is optimal, we require the assumptions of both the impact-minimization robot and the efficient, interactive robot, each of which individually presents complex challenges in implementation.

# 5    Conclusion and Further Work

In this paper, we present a novel model of value (mis)alignment between a human principal and an AI agent. Fundamental to this model is that the human provides an incomplete proxy of their own utility function for the AI agent to optimize. Within this framework, we derive necessary and sufficient theoretical conditions for value misalignment to be arbitrarily costly. Our results dovetail with an emerging literature that connects harms from AI systems to shallow measurement of complex values (32; 20; 5). Taken together, we view this as strong evidence that the ability of AI systems to be useful and beneficial is highly dependent on our ability to manage the fundamental gap between our qualitative goals and their representation in digital systems.

Additionally, we show that abstract representations of techniques currently being explored by other researchers can yield solutions with guaranteed utility improvement. Specifically, impact minimization and human interactivity, if implemented correctly, allow AI agents to provide positive utility in theory. In future work, we hope to generalize the model to account for fundamental preference uncertainty (e.g., as in (12)), limits on human rationality, and the aggregation of multiple human preferences. We are optimistic that research into the properties and limitations of the communication channel within this framework will yield fruitful insights in value alignment, specifically, and AI, generally.

## Broader Impact

As AI systems become more capable in today's society, the consequences of misspecified reward functions increase as well. Instances where the goal of the AI system and the preferences of individuals diverge are starting to emerge in the real world. For example, content recommendation algorithms optimizing for clicks causes clickbait and misinformation to proliferate (33). Our work rigorously defines this general problem and suggests two separate approaches for dealing with incomplete or misspecified reward functions. In particular, we argue that, in the absence of a full description of attributes, the incentives for real-world systems need to be plastic and dynamically adjust based on changes in behavior of the agent and the state of the world.

## Acknowledgments and Disclosure of Funding

This work was partially supported by AFOSR, ONR YIP, NSF CAREER, NSF NRI, and OpenPhil. We thank Anca Dragan, Stuart Russell, and the members of the InterACT lab and CHAI for helpful advice, guidance, and discussions about this project.

## Footnotes

[1]We use "human" and "robot" to refer to any designer and any AI agent, respectively, in our model.

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
