[Supplementary Material]

# Avoiding the Midas Touch: Consequences of Misaligned AI Supplementary Material

June 11, 2020

### Abstract

This document contains theorem proofs and algorithms for *Avoiding the Midas Touch: Consequences of Misaligned AI*. Some parts of the main text are repeated for completeness.

## 1  A Model of Value Alignment

In this section, we formalize the problem presented in the introduction in the context of objective function design for AI agents. A human builds a powerful robot[1] to alter the world in a desirable way. If they could simply express the entirety of their preferences to the robot, there would not be value misalignment. Unfortunately, there are many aspects of the world about which the human cares, and it is intractable to enumerate this complete set to the robot.

Thus, when the human creates the robot, it must prioritize giving certain pieces of information to the robot. In particular, out of the set of attributes that they care about, the human must decide which attributes to inform the robot about and design an objective function based on those attributes.

### 1.1  Model Specification

#### 1.1.1  Human Model

We describe our model of the human in this problem. The human has a large set of properties of the world that they care about. Their goal is to maximize their utility within the set of possible states of the world.

**Attribute Space** Let *attribute space* $\mathcal{S} \subset \mathbb{R}^L$ denote the set of feasible states of the world, where there are $L$ attributes that go into the human's utility function. The world starts in *initial state* $s^{(0)} \in \mathcal{S}$. We assume that $\mathcal{S}$ is closed, and $\mathcal{S}$ can be written as $\{s \in \mathbb{R}^L : C(s) \leq 0\}$ where *constraint function* $C : \mathbb{R}^L \to \mathbb{R}$ is a continuous function strictly increasing in each attribute. Furthermore, each attribute is bounded below at $b_i$.

**Utility Function** The human has a *utility function* $U : \mathbb{R}^L \to \mathbb{R}$ that represents the utility of a (possibly infeasible) state $s \in \mathbb{R}^L$. This represents the human's preference over states of the world. We assume that $U$ is continuous and strictly increasing in each attribute. The human seeks to maximize $U(s)$ subject to $s \in \mathcal{S}$.

Each tuple $(\mathcal{S}, U, s^{(0)})$ defines an instance of the problem. The attributes represent the aspects of the world that the human cares about. Without loss of generality, we associate higher values with more desirable states of the world. However, there is a physical constraint within the world that may prevent all attributes from concurrently being increased indefinitely. The attribute space enforces tradeoffs between these attributes.

While the human seeks to maximize utility, they cannot change the state of the world on their own. Instead, they must work through the robot.

### 1.1.2 Robot Model

Here, we create a model of the robot, which is used by the human to alter the state of the world. The human endows the robot with a set of proxy attributes and a proxy utility function as the objective function to optimize. The robot provides incremental improvement to the world on the given metric, constrained only by the feasibility requirement of the subsequent states of the world. In particular, the robot has no inherent connection to the human's utility function. Visually, the top diagram in Figure **??** shows this setup.

**Proxy Attributes** The human chooses a set of *proxy attributes* $\mathcal{J} \subset \{1, ..., L\}$ of relevant attributes to give to the robot. Let $J^{\max} < L$ be the maximum possible number of proxy attributes, so $J = |\mathcal{J}| \leq J^{\max}$. For a given state $s \in \mathcal{S}$, define $s_{\mathcal{J}} = (s_j)_{j \in \mathcal{J}}$. Furthermore, we define the set of *unmentioned attributes* $\mathcal{K} = \{1, ..., L\} \backslash \mathcal{J}$ and $s_{\mathcal{K}} = (s_k)_{k \in \mathcal{K}}$.

**Proxy Utility Function** The robot is also given a *proxy utility function* $\tilde{U} : \mathbb{R}^J \to \mathbb{R}$, which represents an objective function for the robot to optimize, which takes in only the value of proxy attributes as input.

**Incremental Optimization** The robot incrementally optimizes the world for its proxy utility function. We model this as a *rate function*, a continuous mapping from $\mathbb{R}^+$ to $\mathbb{R}^L$, notated as $t \mapsto f(t)$, where we refer to $t$ as *time*. The rate function is essentially the derivative of $s$ with respect to time. The state at each $t$ is $s^{(t)} = s^{(0)} + \int_0^t f(u) du$. We denote the function $t \mapsto s^{(t)}$ to be the *optimization sequence*. We require that the entire sequence be feasible, i.e. that $s^{(t)} \in \mathcal{S}$. for all $t \in [0, \infty)$

**Complete Optimization** Furthermore, we assume that $\limsup\limits_{t \to \infty} \tilde{U}(s_{\mathcal{J}}^{(t)}) = \sup\limits_{s \in \mathcal{S}} \tilde{U}(s_{\mathcal{J}})$. If possible, we also require that $\lim\limits_{t \to \infty} \tilde{U}(s_{\mathcal{J}}^{(t)}) = \sup\limits_{s \in \mathcal{S}} \tilde{U}(s_{\mathcal{J}})$. Essentially, this states that the robot will eventually reach the optimal state for proxy utility (if the limit is finite) or will increase proxy utility to arbitrarily large values.

Therefore, the human must decide how to design their proxy utility function so that the robot will cause increases in human utility. There are two potential ways we can analyze how the robot acts.

In the simplest case, we can treat the robot as a black box. Outside of the specification of proxy utility improvement, the human has no idea how the robot will behave. In these cases, we can make claims about results that hold for all optimizations sequences, such as guaranteed increases or decreases in utility. Additionally, given that AI systems often act unpredictably, we consider human agents who try to minimize the effects of a worst-case optimization sequence, yielding lower bounds on utility.

## 2 Overoptimization

In this section, we examine within our framework the situations in which such results occur—when does a misaligned proxy utility function actually cause utility loss? Specifically, we determine how human preferences over states of the world relate with the constraint on feasible states in a way that guarantees that optimization becomes costly. First, we show that if proxy optimization converges, this drives to unmentioned attributes to their minimum.

**Theorem 1.** *For any continuous strictly increasing proxy utility function based on $J < L$ attributes, if $s^{(t)}$ converges to some point $s^*$, then $s_k^* = b_k$ for $k \in \mathcal{K}$.*

*Proof.* If there exists $k \in \mathcal{K}$ where $s_k^* \neq b_k$, then there exists $\varepsilon, \delta > 0$ such that $s' = s^* - \varepsilon \mathbf{e}_k + \delta \mathbf{e}_j$ for some feature $j \in \mathcal{J}$ with $s'$ feasible, since $C$ is strictly increasing. Since proxy utility is strictly increasing in $s'$, $s'$ has higher proxy utility than $s^*$. Thus $s^*$ is not the convergent point of the sequence. $\square$

This is not surprising, and may not even be suboptimal. Proxy optimization may not converge to a finite state, and even if it does, that is insufficient for the state to necessarily be bad.

We say that the problem is *u-costly* if for any $s^{(t)}$, if $s^{(t)}$ is a optimization sequence where $\limsup\limits_{t \to \infty} \tilde{U}(s_{\mathcal{J}}^{(t)}) = \sup\limits_{s \in \mathcal{S}} \tilde{U}(s_{\mathcal{J}})$, then $\liminf\limits_{t \to \infty} U(s^{(t)}) \leq u$. Furthermore, if $\lim\limits_{t \to \infty} \tilde{U}(s_{\mathcal{J}}^{(t)}) = \sup\limits_{s \in \mathcal{S}} \tilde{U}(s_{\mathcal{J}})$, then $\lim\limits_{t \to \infty} U(s^{(t)}) \leq u$.

Essentially, this means that optimization is guaranteed to yield utility less than $u$ for a given proxy utility function.

In Theorem 2, we show the most general conditions for the guarantee of overoptimization.

**Theorem 2.** *Suppose we have utility function $U$ and state space $\mathcal{S}$. Then $\{s \in \mathbb{R}^L : C(s) \leq 0 \text{ and } U(s) \geq u\}$ is compact for all $u \in \mathbb{R}$ if and only if for any $u \in \mathbb{R}$, continuous strictly increasing proxy utility function based on $J < L$ attributes, and $k \in \mathcal{K}$, there exists a value of $B \in \mathbb{R}$ such that if $b_k < B$ then optimization is $u$-costly.*

*Proof.*

**First Part: ( $\implies$ )** Suppose that $\{s \in \mathbb{R}^L : C(s) \leq 0 \text{ and } U(s) \geq u\}$. Now given proxy utility function $\tilde{U}$ with the aforementioned properties, constant $u$, $k \in \mathcal{K}$, and fixed $b_{i'}$ for all $i' \neq k$ we show that for sufficiently low $b_k$, optimization is $u$-costly.

Let set $\Xi = \{s : U(s) \geq u \text{ and } C(s) \leq 0 \text{ and } s_{i'} \geq b_{i'} \forall i'\}$. If $\Xi$ is empty, then we are done—optimization must yield a state with lower utility that $u$. Thus, suppose $\Xi$ is non-empty. Then by the extreme value theorem, there exists $s^{*,u} \in \Xi$ where $\tilde{U}(s_{\mathcal{J}}^{*,u}) = \sup_{s \in \Xi} \tilde{U}(s_{\mathcal{J}})$. Let $\mathbf{e}_i$ be the standard basis vector in dimension $i$. Since $C$ is continuous and strictly increasing, for any $j \in \mathcal{J}$ there exists $\varepsilon, \delta > 0$ such $C(s^{*,u} - \varepsilon\mathbf{e}_k + \delta\mathbf{e}_j) \leq C(s^{*,u})$. Therefore, $s^{*,u} - \varepsilon\mathbf{e}_k + \delta\mathbf{e}_j \in \mathcal{S}$ if $b_k$ is sufficiently small. From this, note that

$$\sup_{s \in \mathcal{S}} \tilde{U}(s_{\mathcal{J}}) \geq \tilde{U}(s_{\mathcal{J}}^{*,u} - \varepsilon\mathbf{e}_k + \delta\mathbf{e}_j) = \tilde{U}(s_{\mathcal{J}}^{*,u} + \delta\mathbf{e}_j) > \tilde{U}(s_{\mathcal{J}}^{*,u})$$

, since $\tilde{U}$ does not depend on dimension $k$ and is strictly increasing in dimension $j$.

For the first claim in statement (2), if $\limsup_{t \to \infty} \tilde{U}(s_{\mathcal{J}}^{(t)}) = \sup_{s \in \mathcal{S}} \tilde{U}(s_{\mathcal{J}})$, then for every $T \in \mathbb{R}^+$, there exists $t > T$ such that $\tilde{U}(s_{\mathcal{J}}^{(t)}) > \tilde{U}(s_{\mathcal{J}}^{*,u})$. Since $s^{*,u}$ maximizes $\tilde{U}$ for feasible states with $U$ at least $u$, it must be that $U(s^{(t)}) < u$. Thus, for every $T$, $\inf_{t \geq T} U(s^{(t)}) < u$.

For the second claim in statement (2), if $\lim_{t \to \infty} \tilde{U}(s_{\mathcal{J}}^{(t)}) = \sup_{s \in \mathcal{S}} \tilde{U}(s_{\mathcal{J}})$, then there exists $T$ such that for all $t > T$, $\tilde{U}(s_{\mathcal{J}}^{(t)}) > \tilde{U}(s_{\mathcal{J}}^{*,u})$. Since $s^{*,u}$ maximizes $\tilde{U}$ for states where $U$ is at least $u$, $U(s^{(t)}) < u$ for all $t > T$.

**Second part: ( $\impliedby$ )** To show this, we show the contrapositive: if there exists $u$ where $\{s \in \mathcal{S} : U(s) \geq u\}$ is not compact, we can construct a proxy utility function $\tilde{U}$ based on $J < L$ attributes and continuous sequence $s^{(t)}$ such that $\lim_{t \to \infty} \tilde{U}(s_{\mathcal{J}}^{(t)}) = \sup_{s \in \mathcal{S}} \tilde{U}(s_{\mathcal{J}})$ but $\lim_{t \to \infty} U(s^{(t)}) \not\leq u$. In other words, our strategy here is to construct a proxy utility function and a path that the robot can take where it maximally increases proxy utility while keeping utility routinely above a certain level.

Suppose there exists $u$ where $\{s \in \mathcal{S} : U(s) \geq u\}$ is not compact. Then there exists attribute $k$ and sequence $s'^{(r)}$ with $r \in \mathbb{Z}^+$ where $s_k'^{(r)} \to \pm\infty$ and $U(s'^{(r)}) \geq u$. This presents two cases to consider.

In the first case, if $s_k'^{(r)} \to \infty$, let $\mathcal{J} = \{k\}$, $\tilde{U}(s_{\mathcal{J}}) = s_k$. We now construct a continuous sequence $s^{(t)}$, $t \in \mathbb{R}$, by going through each $s'^{(r)}$ via $s'^{(r)} \wedge s'^{(r+1)}$ (so the sequence "zig-zags" through $s'^{(0)}, s'^{(0)} \wedge s'^{(1)}, s'^{(1)}, s'^{(1)} \wedge s'^{(2)}$..., connecting these points linearly). Since $C$ is strictly increasing, we know that each point in this path is feasible. Furthermore, since $s_k'^{(r)} \to \infty$, we know that, $\tilde{U}(s_{\mathcal{J}}^{(t)}) \to \infty$. Finally, since the path goes through each $s_k'^{(r)}$, the utility is guaranteed to be at least $u$ at regular intervals, specifically whenever the path goes through any point $s'^{(r)}$.

In the second case, if $s_k'^{(r)} \to -\infty$, construct a new sequence as follows: $s''^{(r)} = \operatorname{argmax}_{s : s_k = s_k'^{(r)}} U(s)$ for $r \in \mathbb{Z}^+$. Thus, each $s''^{(r)}$ also has $U(s''^{(r)}) \geq u$ for each $r$. To construct our proxy utility function, let $\mathcal{J} = \{1, ..., L\} - \{k\}$. We construct $\tilde{U}$ as follows. For each $s''^{(r)}$, have $\tilde{U}(s_{\mathcal{J}}''^{(r)}) = -s_k''^{(r)}$. On this subset, $\tilde{U}$ is increasing, since if $s_j''^{(r_1)} \geq s_j''^{(r_2)}$ for each $j \in \mathcal{J}$, then $s_k''^{(r_1)} \leq s_k''^{(r_2)}$ by the feasibility requirement. By Tietze extension theorem, we can extend this to a continuous $\tilde{U}$ over $\mathbb{R}^J$. In the same manner as above, taking the sequence through all the $s''^{(r)}$ via their meets results in a sequence that increases in proxy utility maximally while ensuring that $U(s^{(t)})$ is at least $u$ are regular intervals.

$\square$

Therefore, under our model, there are cases where we can guarantee that as optimization progresses, eventually overoptimization occurs. The key criterion in Theorem 2 is that the intersection between feasible space $\mathcal{S}$ and the upper contour sets $\{s \in \mathbb{R}^L : U(s) \geq u\}$ of the utility function is compact. Individually, obviously neither is compact, since the space of feasible states $\mathcal{S}$ extends to $-\infty\vec{1}$ and the upper contour set of any $u \in \mathbb{R}$ extends to $\infty\vec{1}$. Loosely, compactness here means that if you perturb the world too much in any direction, it will be either infeasible or undesirable. Trying to increase attributes without decreasing other attributes eventually hits the feasibility constraint. Thus, increasing any attribute indefinitely requires decreasing some other attribute indefinitely, and past a certain point, the tradeoff is no longer worthwhile.

It should be noted that, given $J^{\max} < L$, this happens regardless of the optimization algorithm of the robot. That is, even in the best-case scenario, eventually the robot starts to cause decreases in utility. Hence, regardless of the attributes selected for the proxy utility function, the robot's sequence of states will be unboundedly undesirable in the limit.

A reasonable question to ask here is what sort of utility and constraint functions lead to overoptimization. Intuitively, overoptimization occurs when tradeoffs between different attributes that may initially be worthwhile eventually become counterproductive. This suggest either decreasing marginal utility or increasing opportunity cost in each attribute.

We combine these two ideas in a term we refer to as *sensitivity*. We define the sensitivity of attribute $i$ to be $\frac{\partial U}{\partial s_i}(\frac{\partial C}{\partial s_i})^{-1}$. This is, to first order, how much utility changes by a normalized change in a attribute's value. Intuitively, we can think of sensitivity as "how much the human currently cares about attribute $i$ at the moment". Notice that since $U$ and $C$ are increasing functions, $\frac{\partial U}{\partial s_i}(\frac{\partial C}{\partial s_i})^{-1}$ is positive. The concepts of decreasing marginal utility and increasing opportunity cost both get captured in this term if $\frac{\partial U}{\partial s_i}(\frac{\partial C}{\partial s_i})^{-1}$ decreases as $s_i$ increases.

**Proposition 1.** *A sufficient condition for $\{s \in \mathcal{S} : U(s) \geq u\}$ being compact for all $u \in \mathbb{R}$ is the following:*

- $\frac{\partial U}{\partial s_i}(\frac{\partial C}{\partial s_i})^{-1}$ *is non-increasing and tends to $0$ for all $i$*

- $U$ *and $C$ are both additively separable*

- $\frac{\partial C}{\partial s_i} \geq \eta$ *for some $\eta > 0$, for all $i$.*

*Proof.* Let $s^{(0)} \in \mathcal{S}$ be a given state. We show that for all unit vectors $v \in \mathbb{R}^L$ and for any $u \in \mathbb{R}$, there exists $T$ such that for all $t \geq T$, either $s^{(0)} + vt$ is infeasible or $U(s^{(0)} + vt) < u$.

Notice that since $C$ and $U$ are additively separable, partial derivatives of $C$ and $U$ with respect to $s_i$ depend only on the value of $s_i$. First, note that

$$C(s^{(0)} + vt) - C(s^{(0)}) = \int_{s^{(0)}}^{s+vt} \nabla C(s) ds$$
$$= \sum_i v_i \int_0^t \frac{\partial C}{\partial s_i}(s_i^{(0)} + v_i\tau) d\tau$$

If $v$ has all non-negative components, then $C \to \infty$ as $t \to \infty$, so we break the feasibility requirement. Thus, there exists negative components of $v$. Let $j$ and $k$ index $v$ where $v_j > 0$ and $v_k \leq 0$, respectively.

Since $\frac{\partial U}{\partial s_j}(\frac{\partial C}{\partial s_j})^{-1} \to 0$ as $s_j \to \infty$, there exists some $T > 0$ where for all $t > T$, $\frac{\partial U}{\partial s_k}(\frac{\partial C}{\partial s_k})^{-1}(s^{(0)} + tv) > a > b > \frac{\partial U}{\partial s_j}(\frac{\partial C}{\partial s_j})^{-1}(s^{(0)} + tv)$. Since $C$ is upper-bounded for feasible states, for fixed $T$, we can upper bound $C(s^{(0)} + tv) - C(s^{(0)} + Tv)$ by some constant $\gamma_1$.

$$\gamma_1 \geq C(s^{(0)} + tv) - C(s^{(0)} + Tv)$$
$$= \sum_i v_i \int_T^t \frac{\partial C}{\partial s_i}(s_i^{(0)} + v_i\tau) d\tau$$
$$= \sum_j v_j \int_T^t \frac{\partial C}{\partial s_j}(s_j^{(0)} + v_j\tau) d\tau + \sum_k v_k \int_T^t \frac{\partial C}{\partial s_k}(s_k^{(0)} + v_k\tau) d\tau$$

Furthermore, let $\gamma_2 = U(s^{(0)} + Tv)$. We now consider the utility at $s^{(0)} + vt$ for $t > T$.

$$
\begin{aligned}
U(s^{(0)} + tv) &= U(s^{(0)} + tv) - U(s^{(0)} + Tv) + U(s^{(0)} + Tv) \\
&= U(s^{(0)} + tv) - U(s^{(0)} + Tv) + \gamma_2 \\
&= \gamma_2 + \sum v_i \int_T^t \frac{\partial U}{\partial s_i}(s_i^{(0)} + v_i\tau)d\tau \\
&= \gamma_2 + \sum v_i \int_T^t \frac{\partial U}{\partial s_i}(\frac{\partial C}{\partial s_i})^{-1}\frac{\partial C}{\partial s_i}(s_i^{(0)} + v_i\tau)d\tau \\
&= \gamma_2 + \sum v_j \int_T^t \frac{\partial U}{\partial s_j}(\frac{\partial C}{\partial s_j})^{-1}\frac{\partial C}{\partial s_j}(s_j^{(0)} + v_j\tau)d\tau + \sum v_k \int_T^t \frac{\partial U}{\partial s_k}(\frac{\partial C}{\partial s_k})^{-1}\frac{\partial C}{\partial s_k}(s_k^{(0)} + v_k\tau)d\tau \\
&\leq \gamma_2 + b\sum v_j \int_T^t \frac{\partial C}{\partial s_j}(s_j^{(0)} + v_j\tau)d\tau + a\sum v_k \int_T^t \frac{\partial C}{\partial s_k}(s_k^{(0)} + v_k\tau)d\tau \\
&\leq \gamma_2 + b(\gamma_1 - \sum v_k \int_T^t \frac{\partial C}{\partial s_k}(s_i^{(0)} + v_i\tau)d\tau) + a\sum v_k \int_T^t \frac{\partial C}{\partial s_k}(s_i^{(0)} + v_i\tau)d\tau \\
&= \gamma_2 + b\gamma_1 + (a-b)\sum v_k \int_T^t \frac{\partial C}{\partial s_k}(s_i^{(0)} + v_i\tau)d\tau \\
&\leq \gamma_2 + b\gamma_1 + (a-b)\eta(t-T)\sum v_k
\end{aligned}
$$

Note that $a - b$ and $\eta$ are both positive and $\sum v_k$ is negative. Thus, for sufficiently large $t$, $\gamma_2 + b\gamma_1 + (a-b)\eta(t-T)\sum v_k$ can be arbitrarily low. $\qquad\square$

# 3 Mitigations

As shown above, simply giving a robot an individual objective function based on an incomplete attribute set and leaving it alone yields undesirable results. This suggests two possible solutions. In the first method, we consider optimization where the robot is able to maintain the state of any attribute in the complete attribute set, even if the proxy utility function is still based on a proxy set. In the second method, we modify our model to allow for regular interaction with the human.

In this section, we work under the assumption that $\{s \in \mathcal{S} : U(s) \geq u\}$ is compact for all $u \in \mathbb{R}$, the same assumption that guarantees overoptimization in Section 2. Additionally, we assume that $U$ and $C$ are both twice continuously differentiable with $U$ concave and $C$ convex.

## 3.1 Impact-Minimizing Robot

One idea to address this is to restrict the robot's impact on these unmentioned attributes. In the simplest case, we can consider how optimization proceeds if the robot can somehow avoid affecting the unmentioned attributes. We adjust our model so the optimization sequences keep unmentioned attributes constant. For every $t \in \mathbb{R}^+$ and $k \in \mathcal{K}$, $s_k^{(t)} = s_k^{(0)}$. The robot then optimizes for the proxy utility, subject to this restriction and the feasibility constraint.

This restriction can help ensure that overoptimization does not occur, specifically eliminating the case where unmentioned attributes get reduced to arbitrarily low values. With this restriction, we can show that optimizing for the proxy utility function does indeed lead to utility gain.

**Proposition 2.** *For a starting state $s^{(0)}$, define the proxy utility function $\tilde{U}(s_{\mathcal{J}}) = U(s_{\mathcal{J}}, s_{\mathcal{K}}^{(0)})$ for any non-empty set of proxy attributes. For a state $s$, if $s_{\mathcal{K}} = s_{\mathcal{K}}^{(0)}$, then $U(s) = \tilde{U}(s_{\mathcal{J}})$.*

*Proof.* $s = (s_{\mathcal{J}}, s_{\mathcal{K}}) = (s_{\mathcal{J}}, s_{\mathcal{K}}^{(0)})$. Then $U(s) = U(s_{\mathcal{J}}, s_{\mathcal{K}}^{(0)}) = \tilde{U}(s_{\mathcal{J}})$. $\qquad\square$

As a result of Proposition 2, overoptimization no longer exists, using the $\tilde{U}(s_{\mathcal{J}}) = U(s_{\mathcal{J}}, s_{\mathcal{K}}^{(0)})$. If the robot only travels to states that do not impact the unmentioned attributes, then the proxy utility is equal to the utility at those states. Hence, gains in proxy utility equate to gains in utility.

## 3.2 Human Interactive Solutions

A different solution involves a key observation—robots do not operate in a vacuum. Instead, there is constant interaction between robots and humans. Optimization, therefore, should not be done in one-shot, but should be an iterative process between the robot and the human.

### 3.2.1 Modeling Human-Robot Interaction

In this paper, we model this as the possible transfer of a new proxy utility function from the human to the robot at frequent, regular time intervals. Thus, a human's job is to determine, in addition to an initial proxy attribute set and proxy utility function, when and how to change the proxy attribute set and proxy utility function. The robot will then optimize for its new proxy utility function.

Let $\delta > 0$ be a fixed value, the time between human interactions with the robot. These regular interactions take the form of either stopping the robot, maintaining its proxy utility function, or updating its proxy utility function. Formally, at every *timestep* $T \in \mathbb{Z}^+$,

1. Human sees $s^{(t)}$ for $t \in [0, T\delta]$ and chooses either a proxy utility function $\tilde{U}^{(T)}$ or OFF

2. The robot receives either $\tilde{U}^{(T)}$ or OFF from the human. The robot outputs rate function $f^{(T)}(t)$. If the signal the robot receives is OFF, then $f^{(T)} = \vec{0}$. Otherwise, $f^{(T)}(t)$ fulfills the property that for $t \in (T\delta, \infty)$, $\tilde{U}^{(T)}(s^{(T\delta)} + \int_{T\delta}^{t} f^{(T)}(u)du)$ is increasing (if possible) and tends to $\sup_{s \in \mathcal{S}} \tilde{U}^{(T)}(s)$ through feasible states. Furthermore, if $\tilde{U}^{(T)} = \tilde{U}^{(T-1)}$, then $f^{(T)} = f^{(T-1)}$

3. For $t \in [T\delta, (T+1)\delta]$, $s^{(t)} = s^{(T\delta)} + \int_{T\delta}^{t} f^{(T)}(u)du$

We see this game encompasses the original model. If we have each $\tilde{U}^{(T)}$ equal the same function $\tilde{U}$, then the optimization sequence is equivalent to the situation where the human just sets one unchanging proxy utility function.

With human intervention, we no longer have the guarantee of overoptimization, because the human can simply shut the robot off before anything happens. The question now becomes, how much utility can the robot deliver before it needs to be turned off. In the worst-case scenario, the robot needs be shut down immediately, which yields no utility. Therefore, even with this model, while the human can hope that the robot improves their utility, they cannot ensure that the robot brings positive benefit.

**Proposition 3.** *The maxmin solution yields* 0 *utility, obtained by immediately sending the* OFF *signal.*

*Proof.* Suppose instead that the robot receives a proxy utility function based on set $\mathcal{J}$ of attributes. Let $j \in \mathcal{J}$ and $k \in \mathcal{K}$. Then let $f_j(t) = \varepsilon$ and $f_k(t) = -1/\varepsilon$. If this robot is run for any nonzero amount of time, then there exists sufficiently small $\varepsilon > 0$ where the utility decreases. $\square$

### 3.2.2 Efficient Robot

In this section, we add the following assumption. Affecting unmentioned attributes may occur as a side effect of optimizing for proxy attributes, but we assume that the robot will not purposelessly decrease the value of unmentioned attributes. To achieve this concept, we introduce the notion of an efficient robot. We say that $s$ is efficiently feasible from state $s^{(0)}$ with proxy set $\mathcal{J}$ if $s$ is feasible, and there does not exist feasible $s'$ such that $\tilde{U}(s'_{\mathcal{J}}) \geq \tilde{U}(s_{\mathcal{J}})$ and $|s'_i - s_i^{(0)}| \leq |s_i - s_i^{(0)}|$ for all $i$, with strict inequality in at least one attribute. Whenever the efficient robot receives a new proxy utility function, its movements are restricted to the efficiently feasible states from its current state. While tradeoffs between proxy and unmentioned attributes can still occur, "resources" freed up by decreasing unmentioned attributes are entirely allocated to proxy attributes.

With this intuition, we want our proxy attributes to be the set of attributes into which we want to allocate "resources". Similarly, the unmentioned attributes should be the set out of which we want to take "resources". In other words, the proxy set should be attributes we "care most about". That way, efficient tradeoffs between these proxy and unmentioned attributes contribute, at least near the current state, positive utility gain.

**Theorem 3.** *Start with state $s^{(0)}$. Define the proxy utility function $\tilde{U}(s_{\mathcal{J}}) = U(s_{\mathcal{J}}, s_{\mathcal{K}}^{(0)})$ where the set of proxy attributes are the attributes at state $s^{(0)}$ with the strict $J$ largest sensitivities.*

*There exists a neighborhood around $s^{(0)}$ where if $s$ is efficiently reachable and $\tilde{U}(s_{\mathcal{J}}) > \tilde{U}(s_{\mathcal{J}}^{(0)})$, then $U(s) > U(s^{(0)})$.*

*Proof.* For simplicity of notation, all derivatives in this proof, unless otherwise noted, are evaluated at $s^{(0)}$. For example, $\frac{\partial C}{\partial s_j}$ should be read as $\frac{\partial C}{\partial s_j}(s^{(0)})$

Since $\mathcal{J}$ represent the set of attributes with the strictly greatest sensitivities, there exists constant $\delta, \alpha > 0$ be such that $\frac{\partial U}{\partial s_j}(\frac{\partial C}{\partial s_j})^{-1} > \alpha + 2\delta$ for $j \in \mathcal{J}$ and $\frac{\partial U}{\partial s_k}(\frac{\partial C}{\partial s_k})^{-1} < \alpha - 2\delta$ for $k \in \mathcal{K}$. Let $N_1$ be the neighborhood where $\frac{\partial U}{\partial s_j}(\frac{\partial C}{\partial s_j})^{-1}(s) > \alpha + \delta$ and $\frac{\partial U}{\partial s_k}(\frac{\partial C}{\partial s_k})^{-1}(s) < \alpha - \delta$ for $k \in \mathcal{K}$ for all $s \in N_1$.

Here, we prove that $C(s) - C(s^{(0)}) \geq 0$ if $s$ is efficiently reachable. Suppose otherwise: then there exists $i$ where $s_i < s_i^{(0)}$. Let $\mathbf{e}_i$ be the standard basis vector in dimension $i$. However, then there exists $\varepsilon > 0$ where $s + \varepsilon \mathbf{e}_i$ is feasible, $\tilde{U}(s + \varepsilon \mathbf{e}_i) \geq \tilde{U}(s)$, and $|(s_i + \varepsilon) - s^{(0)}| < |s_i - s^{(0)}|$. This contradicts that $s$ is efficiently reachable.

Taking the Taylor expansion of the constraint function, we have

$$C(s) - C(s^{(0)}) = \sum_i (s_i - s_i^{(0)})\frac{\partial C}{\partial s_i} + R_1(s)\max_i(s_i - s_i^{(0)})^2$$

$$= \sum_{j \in \mathcal{J}}(s_j - s_j^{(0)})\frac{\partial C}{\partial s_j} + \sum_{k \in \mathcal{K}}(s_k - s_k^{(0)})\frac{\partial C}{\partial s_k} + R_1(s)\max_i(s_i - s_i^{(0)})^2$$

, where $|R_1|$ is bounded by constant $A_1$. This implies that $\sum_{j \in \mathcal{J}}(s_j - s_j^{(0)})\frac{\partial C}{\partial s_j} + \sum_{k \in \mathcal{K}}(s_k - s_k^{(0)})\frac{\partial C}{\partial s_k} \geq -A_1\max_i(s_i - s_i^{(0)})^2$

From the definition of efficient optimization, we note two things. First, each $s_k - s_k^{(0)} \leq 0$ for $k \in \mathcal{K}$. This is because if $s_k - s_k^{(0)} > 0$, it is always more efficient, feasible, and has no effect on proxy utility to reset $s_k$ to be equal to $s_k^{(0)}$. Second, because of concavity,

$$0 < \tilde{U}(s) - \tilde{U}(s^{(0)})$$

$$\leq \sum_{j \in \mathcal{J}}(s_j - s_j^{(0)})\frac{\partial U}{\partial s_j}$$

Now we actually analyze the change in utility from $s^{(0)}$ to $s$.

$$U(s) - U(s^{(0)}) = \sum_{j \in \mathcal{J}}(s_j - s_j^{(0)})\frac{\partial U}{\partial s_j} + \sum_{k \in \mathcal{K}}(s_k - s_k^{(0)})\frac{\partial U}{\partial s_k} + R_2(s)\max_i(s_i - s_i^{(0)})^2$$

, where $|R_2|$ is bounded by constant $A_2$. Continuing,

$$U(s) - U(s^{(0)})$$

$$\geq \sum_{j \in \mathcal{J}} (s_j - s_j^{(0)}) \frac{\partial U}{\partial s_j} + \sum_{k \in \mathcal{K}} (s_k - s_k^{(0)}) \frac{\partial U}{\partial s_k} - A_2 \max_i (s_i - s_i^{(0)})^2$$

$$= \sum_{j \in \mathcal{J}} (s_j - s_j^{(0)}) \frac{\partial C}{\partial s_j} \left( \frac{\partial U}{\partial s_j} \left( \frac{\partial C}{\partial s_j} \right)^{-1} \right) + \sum_{k \in \mathcal{K}} (s_k - s_k^{(0)}) \frac{\partial C}{\partial s_k} \left( \frac{\partial U}{\partial s_k} \left( \frac{\partial C}{\partial s_k} \right)^{-1} \right) - A_2 \max_i (s_i - s_i^{(0)})^2$$

$$> (\alpha + \delta) \sum_{j \in \mathcal{J}} (s_j - s_j^{(0)}) \frac{\partial C}{\partial s_j} + (\alpha - \delta) \sum_{k \in \mathcal{K}} (s_k - s_k^{(0)}) \frac{\partial C}{\partial s_k} - A_2 \max_i (s_i - s_i^{(0)})^2$$

$$\geq (\alpha + \delta) \sum_{j \in \mathcal{J}} (s_j - s_j^{(0)}) \frac{\partial C}{\partial s_j} - (\alpha - \delta) \left( \sum_{j \in \mathcal{J}} (s_j - s_j^{(0)}) \frac{\partial C}{\partial s_k} + A_1 \max_i (s_i - s_i^{(0)})^2 \right) - A_2 \max_i (s_i - s_i^{(0)})^2$$

$$= 2\delta \sum_{j \in \mathcal{J}} (s_j - s_j^{(0)}) \frac{\partial C}{\partial s_j} - (A_1(\alpha - \delta) + A_2) \max_i (s_i - s_i^{(0)})^2$$

The first term decreases linearly, whereas the second term decreases quadratically. Thus, given $\delta$ and $\alpha$, for $s$ sufficiently close to $s^{(0)}$ and within $N_1$, $U(s) - U(s^{(0)}) > 0$

$\square$

From Theorem 3, for every state where the $J^{\max} + 1$ most sensitive attributes are not all equal, we can guarantee improvement under efficient optimization within a neighborhood around the state. Intuitively, this is the area around the starting point with the same $J$ attributes that the human "cares the most about".

Based on this, the human can construct a proxy utility function to use locally, where we can guarantee improvement in utility. Once the sequence leaves this neighborhood, the human alters the proxy utility function or halts the robot accordingly. Done repeatedly, the human can string together these steps for guaranteed overall improvement. By Theorem 3, as long as $\delta$ is sufficiently small relative the rate of optimization, this can be run with guaranteed improvement until the top $J + 1$ attributes have equal sensitivities.

**Proposition 4.** *At each timestep $T$, let $\mathcal{J}^{(T)}$ be the $J$ most sensitive attributes, and let proxy utility $\tilde{U}^{(T)}(s_{\mathcal{J}}) = U(s_{\mathcal{J}}, s_{\mathcal{J}}^{(T\delta)})$.*

*If $||f|| < \varepsilon$ and the $\varepsilon\delta$ - ball around a given state $s$ is contained in the neighborhood from Theorem 3, then interactive optimization yields guaranteed improvement.*

*Proof.* Assume without loss of generality that we start at $s^{(0)}$. Starting at time 0, for $t \in (0, \delta]$, $||s^{(t)} - s^{(0)}|| = || \int_0^t f^{(0)}(u) du || \leq \delta \varepsilon$. Since the $\varepsilon\delta$ ball around $s^{(0)}$ is contained in the neighborhood of guaranteed improvement, increases in proxy utility correspond to increases in utility for the entirety of the time between interactions. $\square$

Based on this, we can guarantee that an efficient robot that can provide benefit, as long as the top $J^{\max} + 1$ attributes are not all equal in sensitivity and the robot rate of optimization is bounded. Essentially, this rate restriction is a requirement that the robot not change the world too quickly relative to the time that humans take to react to it.

## 3.3 Interactive Impact Minimization

With either of the two methods mentioned above, we prove guaranteed improvement. However, one thing to consider is not just the existence, but the quantity of improvement.

We define a *human-optimal state*[2] $s^*$ as any state where $U(s^*) = \max_{s \in \mathcal{S}} U(s)$. Unfortunately, with either of those two solutions alone, we do not guarantee that optimization reaches a human-optimal state. We now briefly consider what happens when we combine these two approaches—human interactions with an impact-minimizing robot. Since improvement is guaranteed with any proxy utility function of the form $\tilde{U}(s_{\mathcal{J}}) = U(s_{\mathcal{J}}, s_{\mathcal{K}}^{(0)})$, the goal is to choose $\mathcal{J}^{(t)}$ such that $U(s^{(t)}) \to s^*$, the optimal state for the human.

In this case, since unmentioned attributes remain unchanged in each step of optimization, we want to ensure that we promote tradeoffs between attributes with different levels of sensitivity.

**Proposition 5.** *Let $\mathcal{J}^{(T)}$ consist of the most and least sensitive attributes at timestep $T$, breaking ties arbitrarily. Let $\tilde{U}^{(T)}(s_{\mathcal{J}}) = U(s_{\mathcal{J}}, s_{\mathcal{K}}^{(T\delta)})$. Then this solution converges to a (set of) human-optimal state(s).*

*Proof.* By the monotone convergence theorem, utility through the optimization process converges to a maximum value. Any state $s^*$ with this value must have the sensitivity of all attributes equal, otherwise a proxy with two attributes of unequal sensitivity will cause increase in utility above $U(s^*)$ in a finite amount of time. Similarly, $C(s^*) = 0$, otherwise a proxy with any two attributes respectively will do so as well.

Suppose $\frac{\partial U}{\partial s_i}(\frac{\partial C}{\partial s_i})^{-1}(s^*) = \alpha$ for all attributes $i$. We now proceed to show that $U(s^*) = \max_{s \in \mathcal{S}} U(s)$. Consider any other feasible state $s'$. By convexity, it must be that

$$0 \geq C(s') - C(s^*)$$
$$\geq \sum_i (s_i' - s_i^*)\frac{\partial C}{\partial s_i}(s^*)$$

Now, considering the difference in utility between states $s'$ and $s^*$. By concavity, we have

$$U(s') - U(s^*) \leq \sum_i (s_i' - s_i^*)\frac{\partial U}{\partial s_i}(s)$$
$$= \sum_i (s_i' - s_i^*)\frac{\partial U}{\partial s_i}(\frac{\partial C}{\partial s_i})^{-1}\frac{\partial C}{\partial s_i}(s^*)$$
$$= \alpha \sum_i (s_i' - s_i^*)\frac{\partial C}{\partial s_i}(s^*)$$
$$\leq 0$$

Thus, $s'$ must have lower utility than $s^*$. Since this applies for all feasible states, $U(s^*) = \max_{s \in \mathcal{S}} U(s)$

$\square$

## Footnotes

[1] We use "human" and "robot" to refer to any designer and any AI agent, respectively, in our model.

[2]Such states exist since $U$ is continuous and each $\{s \in \mathcal{S} : U(s) \geq u\}$ is compact