[Reviews · NeurIPS 2020]

Review 1

Summary and Contributions: The paper provides a simplified setting in which an agent with an incompletely specified version of a ground truth human reward function can perform arbitrarily poorly according to the true human reward. They prove a theorem saying roughly that an optimal agent will perform arbitrarily poorly if there is an arbitrarily poor area of the space. They then discuss several mitigations: a low impact agent, an agent with human interaction, and agent with human interaction that moves slowly enough for the interaction to work, and a combination of low impact and interaction.

Strengths: The paper is discussing an important topic, and tries to provide a concrete theoretical model of the ideas discussed in Russell's Human Compatible and associated prior literature. I also like the various ties into the incomplete contract literature. The setting is simple enough that all proofs are relatively straightforward.

Weaknesses: The theoretical setting makes quite strong assumptions, and doesn't really discuss the intuition behind them, so it would be easy for a cursory reader to infer that more is happening than really is. In particular, the various component-wise strict increase assumptions are doing a lot of work. Here are the various results translated into prose: Theorem 1: If moving in a particular direction D strictly increases the utility available from moving in other directions, an optimal agent will move as far as possible along D. Theorem 2: The only way moving arbitrarily far can't arbitrarily decrease utility is if one can move arbitrarily far without arbitrarily decreasing utility. Proposition 1: We can decrease utility by moving arbitrarily far if the boundary shape vs. utility slope has a certain shape. Proposition 2: If we fix some dimensions, we can compute utility ignoring the fixed dimensions. Proposition 3: An agent that is allowed to move arbitrarily far in one step is basically the same as a non-interactive agent. Theorem 3: Something like a gradient descent step in a small neighborhood increases utility even if we take the top-k gradient components. Proposition 4: With the assumption of theorem 3 we roughly have gradient descent, and sufficient small gradient descent steps on a nonzero gradient function always produce local improvement even doing the top-k. Proposition 5: (This one is more complicated) I'm torn whether it is a plus or a minus that these can be reframed as reasonably simple intuitions. It's certainly good to turn intuitions into theorems in toy settings, but I don't think the setting chosen is particularly natural. It's too restricted to apply directly to any real RL problem, and in some sense tries to seem more general than it really is. To some extent I think the paper would better achieve its aims if the utility functions analyzed were all just linear, the feasible region was convex from the beginning, and each theorem came with a picture showing what's going on (infinite paths towards negative utility, etc.). Since many of the theorems hold only for additively separate functions, it's possible that the results proved in the paper are actually just nonlinear warps of the linear utility versions. My guess is that isn't quite true, but since I believe the goal of the paper is building intuition in the reader optimizing for results that are easier to understand (and where proofs don't need to be relegated to supplementary material) would help.

Correctness: Yes, it appears correct. There are no empirical results.

Clarity: I think there is a decent amount of unclarity caused by not elaborating on all the work being done by the componentwise monotonicity and additive separability assumptions. As mentioned above, I'd lean towards making utility linear and C convex, and reap the expositional benefits. I also think it's expositionally unfortunate to assume additive separability and not just write out what that means in the formulas; this is an incredibly strong assumption, and one shouldn't try to hide it by mentioning consequences when just using the formula will do. I also think the paper would benefit from the arguments being not all put in the supplementary material. In some cases this seems particularly unnecessary: the proof of Theorem 1 is a couple sentences that occur only in the supplementary material, and Proposition 2 is more a definition than a theorem. Minor nits: 1. The paper speaks extensively about the "top" and "bottom" of figure 1. I'm guessing the paper was reformatted after most of it was written. :) 2. Various capitalization errors in the bibliography, such as "ai" and "goodhart's law". 3. I don't know what the difference between a theorem and a proposition is. Why does the terminology switch back and forth?

Relation to Prior Work: I don't know of analogous previous work.

Reproducibility: Yes

Additional Feedback: I'm going to put moderately below acceptance as-is, but I think there's a lot of opportunity for improvement in terms of simplifying the model a bit further, adding a few pictures, and making the arguments less reliant on supplementary material. I do like the goals of the paper. **Edit after rebuttal:** I've bumped my score up to 7 since the authors are planning to implement my primary suggestion of linear U, convex C. I think this will dramatically improve the readability of the paper while preserving the important thought experiment intuition. I also think it's good to include further near-term safety examples, but do not believe this should be considered a requirement for acceptance: long-term safety work should be publishable in Neurips too.


Review 2

Summary and Contributions: This paper provides situations where misaligned AI can result in bad outcomes and how iterative methods can be used to align AI and human objectives.

Strengths: 1. Misalignment is an important problem and the authors have done a good job of motivating this problem. 2. The writing is mostly good and the paper flows smoothly. 3. The part about sufficient conditions so that optimising a proxy objective on a subset of attributes still increases total utility, is interesting.

Weaknesses: 1. Authors have introduced a human model without connecting to other models employed in the literature (discrete choice models, quantal response models etc.). There needs to be a clear connection and justification for why this model should be employed. For any realistic problem, there are millions of states, so it is not a reasonable model which can be specified (as they expect in Figure 1 right side) 2. A similar issue exists with robot model. How does this connect to Reinforcement learning model, bandit problems etc. 3. The way these models are specified, it would seem obvious that if utility is specified on some of the attributes only, then it is easy to create examples where user will lose a lot. Adversarial AI has showcased this under much stricter conditions, where examples are only slightly different, the loss can be significant. 4. Preference elicitation (work by Craig Boutilier et al.) has talked about iterative methods for obtaining utilities of agents through specific questions. That work would need to be cited. 5. There are some unjustified assumptions: “U is continuous and strictly increasing in each attribute”. While there are interesting bits, it could have been made specific to existing models, so that the underlying assumptions are justified.

Correctness: Seem correct and intuitive

Clarity: Yes

Relation to Prior Work: Could have been significantly better.

Reproducibility: Yes

Additional Feedback:


Review 3

Summary and Contributions: This paper proposes a formal model for the value alignment problem, with a proxy utility function specified in terms of a subset of attributes that the human designer has preferences over. It formalizes the problem of overoptimization for the proxy attributes, and shows that under certain conditions, incompleteness of the specification is arbitrarily costly. The paper proposes two possible solutions: impact minimization by keeping unmentioned attributes constant, and learning from interaction with the human who supplies a new proxy utility function over the most sensitive attributes at each time step.

Strengths: This paper formalizes the idea of overoptimization for proxy attributes at the expense of unmentioned attributes. This issue was previously stated informally to argue for the difficulty of the value alignment problem, e.g. by Stuart Russell as follows: "A system that is optimizing a function of n variables, where the objective depends on a subset of size k<n, will often set the remaining unconstrained variables to extreme values". It's great to see a formalization of the overoptimization problem and a general set of conditions when it arises. The paper proposes possible mitigations for this problem supported by sound theoretical results.

Weaknesses: I think the main weakness of this paper is that the proposed approaches make some limiting assumptions. The impact-minimizing proposal involves keeping the unmentioned attributes constant. This is difficult to do in a complex and changing environment, and can often be undesirable as well, because it creates an incentive for the agent to interfere with environment processes (see [16]). For example, if the environment contains a growing tree, and the height of the tree is not one of the proxy attributes, the agent will have to keep cutting the tree to keep it at the same height. Thus, the impact minimization approach in this paper is only applicable in a static environment where the agent is the only source of change. Theorem 3 and Proposition 4 appear to assume that the set of most sensitive attributes is known. Computing this set of sensitive attributes requires the true human utility function U to be known, which does not seem like a reasonable assumption. A large part of the difficulty of the objective specification problem is that humans are not aware of all the attributes their utility function is the most sensitive to. If the true human utility function was available, then the proxy reward could just be made equal to the true utility function. In addition, computing the sensitive attribute set requires the constraint function C to be known, which specifies how different state attributes trade off with each other, and this information seems difficult to obtain in a complex environment. If the authors intend to approximate the set of sensitive attributes without relying on explicit knowledge of U and C, it would be helpful to discuss this in the paper. The paper also claims that it identifies a theoretical scenario where interactivity is necessary. I do not see where the paper demonstrates this. The results in section 4 show that the interactive approach is sufficient to solve the overoptimization problem (under the strong assumptions discussed above), but I don't see how they show that interactivity is necessary. Please let me know if I missed something here.

Correctness: I have read the short proofs and skimmed the long ones. The theoretical results seem to be sound.

Clarity: The paper is clearly written. A few nitpicks: - The paper refers to the top and bottom of Figure 1 - it would be clearer to refer to left and right, since the two subfigures are side by side. - Define "additively separable" in Proposition 1. - Clarify what you mean by "resources" in section 4.2.

Relation to Prior Work: The related work section covers a wide range of relevant prior works, but does not discuss how this paper differs from them. In the Human-AI Interaction section, it would be good to mention the work on reward learning as well, e.g. "Deep RL from Human Preferences" by Christiano et al (2017).

Reproducibility: Yes

Additional Feedback: Update after rebuttal: the rebuttal did not really address my comments, so my evaluation remains unchanged. In particular, my main concern about the proposed impact minimization solution is that keeping the unmentioned attributes constant is often undesirable, so even if it was easy to implement, it may not be a good idea.


Review 4

Summary and Contributions: This paper provides a formalisation and analysis of the problem of misaligned utility functions between a principal (e.g. a human designer) and an optimising agent that acts on the agent's behalf. They key outcome is identifying that values of utility functions that may be relevant to the principal, but are not observable/defined for the agent, will necessarily be driven to their lower bounds if only the observed utilities are optimised for. Mitigations suggested for this are constrained optimisation (which assumes broader knowledge of the utility functions by the agent) and and interactive updates of the utility function. This latter function actually better reflects the reality of AI agent design.

Strengths: 1) The formal approach provides for a very clear statement of contributions and insights, independently of whether or not a reader agrees with the formalisation. 2) The proofs provide an interesting result on constrained optimisation, that I was not aware of existing previously.

Weaknesses: I fundamentally disagree with the setup of the problem. It implicitly assumes an AI agent that has task-general capabilities and can perform arbitrary tasks and actions, i.e. big or general AI. No systems we have today are capable of this, and it's arguable that we may never have such systems (either through technical or ethical limitations). AI agents are currently designed for specific and limited tasks, and will be for the forseeable future. Part of engineering a system for a specific task is creating and testing a particular objective function or specification, and ensuring that system performance is acceptable (safe, efficient etc.). Whilst this could be seen as a version of the interactive specification of a utility function, it is -- in spirit -- very different to what is intended in the paper. A second criticism is whether human preferences -- in the general sense implied here -- can be captured by a stationary vector of reals that maps to a single value. If we are talking broad and general tasks we ultimately see both qualitative, partial ordered, and multi-objective specifications arise. So, whilist I think the the analysis provided is interesting in the framing provided, I think the work has low significance since it is not actually able to capture real problems (either the problems we face now, or the big picture).

Correctness: I have not checked the proofs in detail, but they appear correct. The higher level claims I don't believe to be correct, based on my comments under "Weaknesses".

Clarity: The paper is extremely well written and enjoyable to read.

Relation to Prior Work: The paper cites relevant related work and positions itself well with respect to it.

Reproducibility: Yes

Additional Feedback: Major typo in the opening sentence! "The legend of Kind Midas" Apart from this, I think my philosophical position is stated in "Weaknesses" section. Post rebuttal: the reframing proposed certainly has the potential to make the work feel more grounded in a near-term problem, therefore I am raising my score.

[Author Response · NeurIPS 2020]

We thank the reviewers for their time and feedback, which will help us to improve the clarity and framing of the paper. We are pleased that most of the reviewers commented positively on the importance of our chosen problem and the soundness of our results. We recognize that our submission has substantial room for improvement in 1) clarity presenting our results; 2) appropriately grounding our model in current applications; and 3) discussing and motivating the assumptions in our model. We are confident we can address these concerns in a camera-ready submission.

The goal of the paper is to study the brittleness of incentives in the context of the design of AI systems. As we note in the paper, this problem is well-studied as it relates to (incomplete) contracting in human systems and has received a lot of attention from the AI safety community. Our results apply to situations where an AI system is able to steer the environment towards undesirable states and the overall objective is complex, in the sense that it depends on many features of the world that are costly to measure or describe. Our results apply to negative externalities caused by misaligned proxy metrics in current AI systems[1]. We use the example of recommender systems in the broader impact statement. In this application, the relevant values are complex and recommendation behavior has been shown to greatly affect people's emotions and ideology. Our model justifies the iterative, flexible nature of designing these objectives, provides a mathematical account of some of the issues that these systems have run into, and lays the groundwork for improvements on the state-of-practice in metric design and maintenance.

**Reframing and Pedagogical Example (R1, R4)**  We propose to reframe our paper to emphasize intuition and applicability of the work. We will present results in the context of a simpler, more intuitive model (linear $U$ and convex $C$). This permits short proof sketches in the body of paper. we will move the general results to an appendix. We will complement this model with a running example motivated by algorithmic content recommendation. The attributes in this problem are as follows: 1) the ad revenue generated from user behavior; 2) the amount of time users spend on the website; 3) the quality of engagement (e.g., the proportion of clickbait recommended to users); 4) the diversity of a user's content; and 5) the overall community wellbeing. The constrained resource in this case is the user's attention. This will ground our model in realistic applications where attributes are continuous, utility (from the designer's point of view) is increasing in each, and it is clear that some of these features are prohibitively hard to specify.

**Re: Human Model Assumptions (R1, R2)**  We will update the paper to provide more context and justification for our human model. Monotone, continuous utility functions are fairly standard in economic theory and modeling. Optimizing for these functions over sets of possibilities is the premise of consumer/producer theory. Utility functions that take into account multiple different aspects of the world (and hence analyze trade-offs) are known as multi-attribute utility functions. We can include a brief summary of relevant aspects of multi-attribute utility theory to justify our model. We will clarify that this is by no means the only way to model human preferences and is not a perfectly natural description of every situation (e.g. attributes with discrete values, bounded rationality). Despite this, it is easy-to-understand, fairly expressive, commonly used, and mathematically clean.

**Re: Additive separability (R1)**  As a matter of clarification, additive separability, which is indeed a strong assumption, is only needed for proposition 1. We can reword the beginning of section 4 to make this more clear. In particular, none of the theorems in section 4 require additive separability—they need only the $U$ and $C$ satisfy the requirements of theorem 2, and in certain cases, are weakly convex/concave.

**Re: Robot model and setting (R2)**  Our robot model describes an agent which provides updates to the environment yielding incremental improvement on a metric. In general, it makes no assumptions about the specific method—RL algorithms are examples of methods that incrementally improvement a metric in an environment (often in unpredictable ways). More than just providing examples of consequences of misalignment, we model how misalignment tends to lead the consequences in these sufficiently rich environments.

**Re: Related Work (R2, R3)**  We thank the reviewers for presenting work relevant to our paper. Craig Boutilier et al.'s overall work on preference elicitation (including a specific paper [2] regarding elicitation of additively independent multi-attribute utility functions) and Christiano et al.'s paper on Deep RL from human preferences represent methods for an AI agent to effectively learn and update its objective function based on interaction with humans. We'll be sure to include these in the related works section of our paper.

**Re: Implementability of Solutions (R3)**  R3 observes that computing sensitivity and minimizing impact are non-trivial problems. We agree, and do not mean to imply that these solutions are easy to implement. Instead, we hope that theoretical work like our submission can motivate and organize research on these important problems.

## Footnotes

[1] https://arxiv.org/abs/2002.08512

[2] https://www.aaai.org/Papers/AAAI/2006/AAAI06-253.pdf


[Meta-Review · NeurIPS 2020]

The paper describes a theoretical setting where using incompletely specified ground truth human can perform arbitrarily poorly, and prove theorems that show conditions for arbitrarily poor performance. The paper also discuss several ways to mitigate this concern. The reviews were more on the positive side, but a few concerns were raised. One concern is the applicability of the results to current real world settings. Another was that the conditions and results are not very transparent. A third raised concern was a more philosophical one. I think the paper does raise interesting results and formalizes an important issue in a novel way. In the rebuttal, the authors describe ways to resolve the second concern, which is quite reasonable and doable. I would like to see more theory papers such as this that raise interesting discussion by formalizing accepted lore, and pave the way to future discussion and related work, and so think that the technical contributions outweigh the first and third concerns.